# INFORMATION-ORDERED BOTTLENECKS FOR ADAPTIVE DIMENSIONALITY REDUCTION

## ABSTRACT

We present the information-ordered bottleneck (IOB), a neural layer designed to adaptively compress data into latent variables ordered by likelihood maximization. Without retraining, IOB nodes can be truncated at any bottleneck width, capturing the most crucial information in the first latent variables. Unifying several previous approaches, we show that IOBs achieve near-optimal compression for a given encoding architecture and can assign ordering to latent signals in a manner that is semantically meaningful. IOBs demonstrate a remarkable ability to compress embeddings of image and text data, leveraging the performance of SOTA architectures such as CNNs, transformers, and diffusion models. Moreover, we introduce a novel theory for estimating global intrinsic dimensionality with IOBs and show that they recover SOTA dimensionality estimates for complex synthetic data. Furthermore, we showcase the utility of these models for exploratory analysis through applications on heterogeneous datasets, enabling computer-aided discovery of dataset complexity.

## 1 INTRODUCTION

Modern deep neural networks (DNNs; LeCun et al., 2015) excel at discovering complex signals and constructing informative high-level latent representations. However, their complexity often leads to computational burden, memory requirements, lack of interpretability, and potential overfitting. Latent compression, which reduces the dimensionality of a network's latent space while preserving essential information, has emerged as a solution to these challenges. In this paper, we propose a generic method for adaptively compressing and ordering the latent space of any DNN architecture, with applications in data exploration and model interpretability.

Classical linear methods, such as Principle Component Analysis (PCA; Jolliffe & Cadima, 2016), fail when dealing with nonlinearly correlated features in datasets. Nonlinear extensions like kernel PCA (Schölkopf et al., 2005) offer some improvements, but they often struggle or become intractable in high-dimensional data. However, deep autoencoders have demonstrated remarkable capabilities in fitting nonlinear manifolds, even in high-dimensional settings (Kingma & Welling, 2013; Gondara, 2016; He et al., 2022). Despite their frequentist training procedure, these autoencoders have theoretical foundations providing Bayesian guarantees on their expressibility (Tishby et al., 2001; Alemi et al., 2016; Saxe et al., 2019). Moreover, the rise of multimodal models and zero-shot inference (Ramesh et al., 2021; 2022) has sparked interest in creating and understanding semantic latent spaces in DNNs (Radford et al., 2021).

The goal of our approach is to embed latent information in a neural network such that, at inference time, we can dynamically vary the bottleneck width while ensuring likelihood maximization. We achieve this by providing a loss-based incentive for a neural network to prioritize certain neural pathways over others. The secondary effects of this are that we can study the inference performance of our model as a function of bottleneck width and interpret which data features carry the most information.

The contributions of this work are as follows: We introduce the information-ordered bottleneck (IOB), a neural layer that compresses high-level data features and orders them based on their impact on likelihood maximization, unifying previous approaches under a single framework. Next, through autoencoding experiments on synthetic and real datasets, our IOB models achieve optimal data compression for a given neural architecture while capturing semantic properties of the data distribution.

Lastly, we propose a novel methodology for estimating intrinsic dimensionality (ID) using IOBs, achieving SOTA performance on high-dimensional synthetic experiments.

## 2 RELATED WORK

The concept and application of IOB layers connect with various machine learning fields: ordered component analysis, latent disentanglement, and intrinsic dimensionality estimation. Ordered component analysis seeks to describe and rank-order latent representations by their impact on likelihood maximization. Classical linear (Jolliffe & Cadima, 2016) or kernel-based methods (Schölkopf et al., 2005) often fail in the limits of high-dimensional data spaces. In deep neural networks, autoencoding frameworks are very effective at dimensionality reduction but do not inherently assign rank-ordered importance to its learned features. To address this problem, Rippel et al. (2014) introduced Nested Dropout, a method by which one randomly truncates the width of a hidden layer during training to encourage a latent ordering in autoencoders. In the case of linear activation functions, Nested Dropout exactly recovers the PCA solution. Independently, Staley & Markowitz (2022) introduced Triangular Dropout, which deterministically weighted the loss function of an autoencoder using reconstructions from every possible configuration of truncated hidden layers. This marginalization over truncation was computed exactly, at every training step, instead of stochastically as in Rippel et al. (2014). In subsequent sections, we show that these two methods are special cases of IOBs, subject to certain hyperparameter choices, and compare them empirically under the same experiments. Lastly, our IOB approach also bears resemblance to recent methods which place autoregressive priors on latent representations (Minnen et al., 2018). These models incorporate autoregressive hyperpriors on the decoder, enhancing reconstruction by contextual information from prior decoded latents. However, they differ fundamentally from the IOB approach in that latent hyperpriors involve a separate hyper autoencoder, while IOBs are part of a single architecture.

Latent disentanglement (Mathieu et al., 2019) is the practice of decomposing the latent space of an autoencoder into explicitly independent components. Methods such as $\beta$-VAE (Higgins et al., 2016) achieve this by imposing a constraint on the representation layer to minimize the mutual information between different latent nodes. This allows for regularization during training, improved model interpretability, and latent interpolation at inference time. Later, the Principle Component Analysis Autoencoder (PCAAE; Pham et al., 2022) combined ordered representation and latent disentanglement through computing latent embeddings by learning one latent variable at a time with separate encoders. While latent disentanglement is not strictly enforced for IOBs, we show that the IOB framework can also be configured to encompass PCAAE training as well.

Intrinsic dimensionality (ID; Ansuini et al., 2019) estimation is used to determine the minimum number of meaningful dimensions needed to capture the essential structure of a dataset. ID estimates can be used to interpret an unknown dataset or to set hyperpriors on the widths of neural networks. ID estimators generally rely on the assumption that samples in a local region of the data manifold are distributed uniformly, allowing the use of nearest neighbor metrics. However, these approaches scale extremely poorly to high-dimensional data (Horvat & Pfister, 2022). Recently, ID estimators utilizing generative neural networks (e.g. Tempczyk et al., 2022; Horvat & Pfister, 2022; Bonheme & Grzes, 2022) have made significant progress towards capturing the high-dimensional regime. To the authors' knowledge, the IOB approach is unique from these works in that it is the first application of an ordered autoencoder method for estimating ID.

## 3 INFORMATION-ORDERED BOTTLENECKS

Consider a dataset of $N$ input-output pairs $\mathcal{D} := \{(\mathbf{x}_i, \mathbf{y}_i)\}_{i=1}^{N}$ for inputs $\mathbf{x} \in \mathbb{X}$ and outputs $\mathbf{y} \in \mathbb{Y}$. We assume there exists an optimal relationship $f^* : \mathbb{X} \to \mathbb{Y}$ which maximizes a given joint log-likelihood $\ell : (\mathbb{X}, \mathbb{Y}) \to \mathbb{R}$ for all $\mathbf{x} \in \mathbb{X}$ and $\mathbf{y} \in \mathbb{Y}$. In this application, we attempt to learn a mapping $\mathbb{X} \to \mathbb{Y}$ through the composition of two parametric transformations $e_\phi : \mathbb{X} \to \mathbb{Z}$ and $d_\eta : \mathbb{Z} \to \mathbb{Y}$, which respectively map inputs $\mathbf{x}$ to latent representations $\mathbf{z} \in \mathbb{Z}$ and, subsequently, latent representations $\mathbf{z}$ to outputs $\mathbf{y}$. We define the composition of these functions as $f_\theta := d_\eta \circ e_\phi$

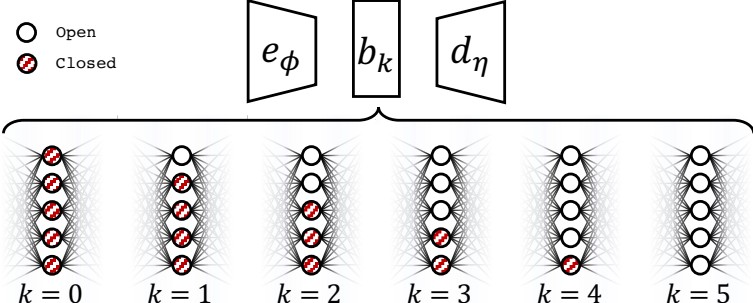

Figure 1: Example concept diagram of IOBs where $k_{\max} = 5$. During each training step, the bottleneck width $k$ is varied via masking. The model is optimized with the summative loss in Equation 2. This results in a model which is incentivized to pass as much information as possible through the top nodes, which are the most reliable during training.

described by a joint parameter vector $\theta := \{\phi, \eta\}$. In this context, our goal is then to compute:

$$\hat{\theta} = \arg\max_{\theta \in \Theta} \sum_{i=1}^{N} \ell \left[ f_\theta \left( \mathbf{x}_i \right), \mathbf{y}_i \right]. \tag{1}$$

This formulation is generalizable to various machine learning problems, including regression, classification, and autoencoding or variational density estimation.

This problem is considered a bottleneck if the dimensionality of the latent representation space is smaller than that of either the input or output space. i.e. $\dim(\mathbb{Z}) < \dim(\mathbb{X})$ or $\dim(\mathbb{Z}) < \dim(\mathbb{Y})$. For sufficiently flexible, non-linear $e_\phi$ and $d_\eta$, such as deep neural networks, it is theoretically possible to compress any distribution in $\mathbb{X}$ into $\mathbb{Z}$ where $0 < \dim(\mathbb{Z}) < \dim(\mathbb{X})$. In fact, both analytic and empirical evidence suggests that the introduction of a bottleneck actually improves the training procedure of complex problems (Saxe et al., 2019). However, the dimensionality of $\mathbb{Z}$ is often taken to be a hyperparameter, with trade offs for fitting accuracy versus constraining power (Goldfeld & Polyanskiy, 2020). For the remainder of this work, we will consider $e_\phi$ and $d_\eta$ to be deep neural networks where $\phi$ and $\eta$ are the tunable weights and biases of each neural layer.

Our approach is to implement a training loss which maximizes Equation 1 for all possible bottleneck dimensionalities. First, we introduce an IOB layer $b_k : \mathbb{Z} \to \mathbb{Z}$ designed to dynamically mask out information flowing through our latent representations $\mathbf{z}$ to form an effective bottleneck of dimensionality $k$. The concept is represented graphically in Figure 1. For a given $k$, $b_k$ functions by assigning the output of every $i$-th node equal to 0 for all $i > k$. These 'closed' nodes thus can neither pass any information on to subsequent layers nor propagate gradients back to previous layers. The order in which we open nodes with increasing $k$ is fixed, such that if the $i$-th node is open, then all nodes $j < i$ must also be open. Next, we use this IOB $b_k$ to construct a bottlenecked neural network using our $e_\phi$ and $d_\eta$. The composition of these three transformations, $f_\theta^{(k)} := d_\eta \circ b_k \circ e_\phi : \mathbb{X} \to \mathbb{Y}$, is now an input-output mapping with an adjustable latent dimensionality $\dim(\mathbb{Z}) = k$. Finally, we introduce an extension of Equation 1 which maximizes the log-likelihood over all bottlenecks:

$$\hat{\theta} = \arg\max_{\theta \in \Theta} \sum_{i=1}^{N} \sum_{k=0}^{k_{\max}} \rho_k \ell \left[ f_\theta^{(k)} \left( \mathbf{x}_i \right), \mathbf{y}_i \right]. \tag{2}$$

This new objective function is now a linear combination of the log-likelihood of the model with each bottleneck size $k$, weighted by scalars $\rho_k$ and taken up to a maximum $k_{\max}$, which can be set as $k_{\max} = \min(\dim(\mathbb{X}), \dim(\mathbb{Y}))$ without loss of generality.

The motivation behind this design is to train a neural network to simultaneously learn signal features from data and compress the most important ones into as low-dimensional space as it can manage. In Figure 1, the top $k = 1$ latent variable is the most reliable pathway, as it is open in $k - 1/k$ terms of the loss function. This encourages the network to prioritize passing information through the top node, leveraging architectural flexibility. The resulting latent layer exhibits an ordered arrangement of components based on their contribution to the log-likelihood, with more informative nodes at

low $k$. During inference, incremental opening of the bottleneck allows for improvements in signal modeling and gains in the likelihood. This approach supports model interpretability (Section 7) and ID analysis in autoencoding (Section 6). This adaptive compressor design does not enforce a fixed bottleneck width or strict latent disentanglement (e.g. Pham et al., 2022). Instead, these properties are learned directly from the data, motivating the neural network to efficiently compress the data by maximizing gains in the likelihood through low $k$.

The choices of $\rho_k$'s and the method for computing the summation over $k$ in Equation 2 are hyperparameters that allow us to generalize this framework to previous approaches. An example of this framework can be seen in the implementation of Nested Dropout (Rippel et al., 2014), where $\rho_k$ values follow a geometric distribution with success probability $p \in [0, 1]$. The summation over $k$ is stochastically calculated in each training batch. To address the vanishing gradient problem observed at high $k$, the Rippel et al. (2014) introduced a 'unit-sweeping' procedure, which freezes the learned latents at low $k$ after convergence, leveraging the memoryless property of the geometric distribution with high $p$ to reduce the optimization's condition number. In the limit where $p \approx 1$ and the encoder is sufficiently expressive, Nested Dropout with unit-sweeping is equivalent to PCAAE (Pham et al., 2022), an approach in which a separate encoder is trained to learn one latent at a time, enforcing strict latent disentanglement. Lastly, another implementation, Triangular Dropout (Staley & Markowitz, 2022), uses a constant $\rho_k$ for all $k$ and computes the sum over $k$ exactly in each training step. We implement and compare these methods, examining their properties in practical applications.

We implement two forms of IOBs, hereafter referred to as Linear IOB and Geometric IOB. The former uses a constant weighting in Equation 2, i.e. $\rho_k = \rho_0 \forall k$. The latter uses a geometric distribution of $\rho_k$ such that $\rho_k = (1 - p)^k p$ where $p$ is a hyperparameter. This hyperparameter is fit dependent on the experiment and is listed in Appendix C. In addition, for the Geometric IOB we implement the stochastic loss summation and unit sweeping procedure, following Rippel et al. (2014). We found that, without this procedure, the vanishing gradient problem due to the geometric weighting is too strong to make any training progress.

## 4 EXPERIMENTS

We describe several synthetic and real datasets which are used throughout this work to experiment with IOBs. All models implemented in this study use an autoencoder setting, i.e. with $\mathbb{Y} = \mathbb{X}$, though we suggest extensions to further inference problems as future work. Additional implementation details are provided in Appendix C.

The first dataset is a commonly-used toy example for manifold learning. We sample a set of $N = 10,000$ data points from a noisy S-curve distribution in $\mathbb{R}^3$. The S-curve distribution is a curved 2D manifold in 3D space. We also add a small amount of isotropic Gaussian noise to each sample with a standard deviation $0.1$. For reproducibility, we use the `sklearn.datasets` implementation of the S-curve distribution to sample these points. The 3D and projected distributions of these sampled datapoints are shown in Figure 2a. The autoencoder architecture for this dataset is a feed-forward neural network with three dense layers in each of the encoder $e_\phi$ and decoder $d_\eta$ and a bottleneck of max width $k_{\max} = 4$.

We next define a series of complex synthetic datasets for which the intrinsic dimensionalities are known a priori, termed $n$-Disk. We define a generator which produces single-channel 2D images of size $32 \times 32$, each containing $n$ 'disks' of different sizes and at different positions. Examples of samples from this generator are shown in Figures 2b and 4b, and an explicit algorithm for this generator is described in Algorithm 1 in Appendix C. We generate four separate $N = 10,000$ datasets for each case of 1-, 2-, 3-, and 4-Disk images. We also construct a heterogenous dataset of $N = 10,000$ generated images in which the number of disks is uniformly sampled from $n \in \{1, 2, 3, 4\}$. This latter set introduces the interesting case in which two disks may entirely overlap and thus reduce the effective $n$ by one, a behavior which is explored in Section 7. When encoding this dataset, we use a CNN autoencoder with three convolutional layers and three dense layers in each of the encoder and decoder. The maximum width of the bottleneck is $k_{\max} = 16$.

We also include the standardized dSprite dataset (Higgins et al., 2016; Matthey et al., 2017) as a validation metric for ID estimates in Section 6. dSprite is a dataset of $700,000$ 2D images each of size $64 \times 64$ and containing a sprite icon. The icons are procedurally generated from five latent fac-

tors: shape, scale, orientation, and x/y position. We use a CNN autoencoder with four convolutional layers and three dense layers in the encoder and decoder for compression. The maximum width of the bottleneck is $k_{\max} = 8$.

Lastly, we demonstrate the ability of IOBs to compress latent representations of SOTA zero-shot image-to-image and text-to-image models. Specifically, we use IOBs to compress the distribution of the MS-COCO 2017 Captioning dataset (Lin et al., 2014) in the Contrastive Language-Image Pre-Training (CLIP) embedding space (Radford et al., 2021). In this fashion, we seek to study how the IOBs handle, compress, and order information in semantic embeddings from text and image data. Transforming the full suite of MS-COCO images into CLIP embeddings produces $\sim 100,000$ datapoints in $\mathbf{x} \in \mathbb{R}^{768}$. We then use a deep feed-forward autoencoder with three-layer encoders and decoders with a maximum bottleneck width of $k_{\max} = 384$. Several examples of MS-COCO prompts and images are shown in Figure 2, and more are given in Figures 5 and 6 in Appendix B.

We evaluate the compression quality of our IOB autoencoders using the image-to-image and text-to-image generative diffusion pipelines from unCLIP (Ramesh et al., 2022). In our tests, we transform MS-COCO images and captions into the CLIP semantic space using the pre-trained CLIP encoder and unCLIP diffusion prior, compress and reconstruct them with our latent autoencoders, and use the outputs to condition the unCLIP generative image model to create samples which look semantically similar to the MS-COCO prompts and images. We use the standard unCLIP diffusion prior to map prompts to CLIP embeddings and a finetuned version of Stable Diffusion 2.1 (Rombach et al., 2022) for generating images.

## 5 DIMENSIONALITY REDUCTION

In this section, we empirically test the ability of IOBs to adaptively compress the datasets described in Section 4. We quantitatively and qualitatively compare these compressions to those of several traditional and machine learning baselines for ordered encoding.

### 5.1 BASELINES

We use PCA, kernel PCA, and PCAAE as benchmarks for dimensionality reduction. PCA (Jolliffe & Cadima, 2016) is a general linear embedding strategy which finds a transformation of the data vector which maximizes the data variance at each cardinal axis. Kernel PCA (Schölkopf et al., 2005) is an extension of PCA designed to capture non-linear data by employing a kernel function. However, the inverse transformation of kernel PCA requires the use of an approximate solver and is prone to error. Lastly, we include a SOTA neural network ordered encoding scheme, PCAAE (Pham et al., 2022). PCAAE computes embeddings by learning one latent variable at a time with separate encoders. In our PCAAE implementations, we use the same autoencoding architectures as were used for the IOB, but we train each successive encoder/decoder under the PCAAE procedure. This is an extremely computationally expensive process and could not be computed for the 384-dimensional latent space of the MS-COCO autoencoder (requiring $\sim 1000$ serial GPU-hours).

We also train a suite of normal autoencoders at many different bottleneck widths. These serve as a benchmark for the compression ability of our autoencoder architectures when we relax the disentanglement constraint and do not enforce an adaptive ordered bottleneck. Without these constraints, the separate normal autoencoders should be able to explain more variability in our dataset than those with IOBs, for a given bottleneck size.

### 5.2 RESULTS

Figure 3 demonstrates the qualitative performance of the Linear IOB model on the S-curve, 2-Disk, and MS-COCO experiments. The reconstructions are plotted relative to an equivalent reconstruction with PCA, as the number of open bottlenecks at inference time increases. We observe a clear growth in information content as the bottleneck widens. Qualitatively, the reconstructions exhibit semantic improvements, with underlying concepts gradually appearing as sufficient information is available. In the 2-Disk example, the reconstructions reveal one disk at a time, completing at around $k = 3$ and $k = 6$ and contrasting with the mean field reconstruction of PCA. In the MS-COCO experiment, large-scale features are reconstructed first (e.g., object types, camera conditions) followed by

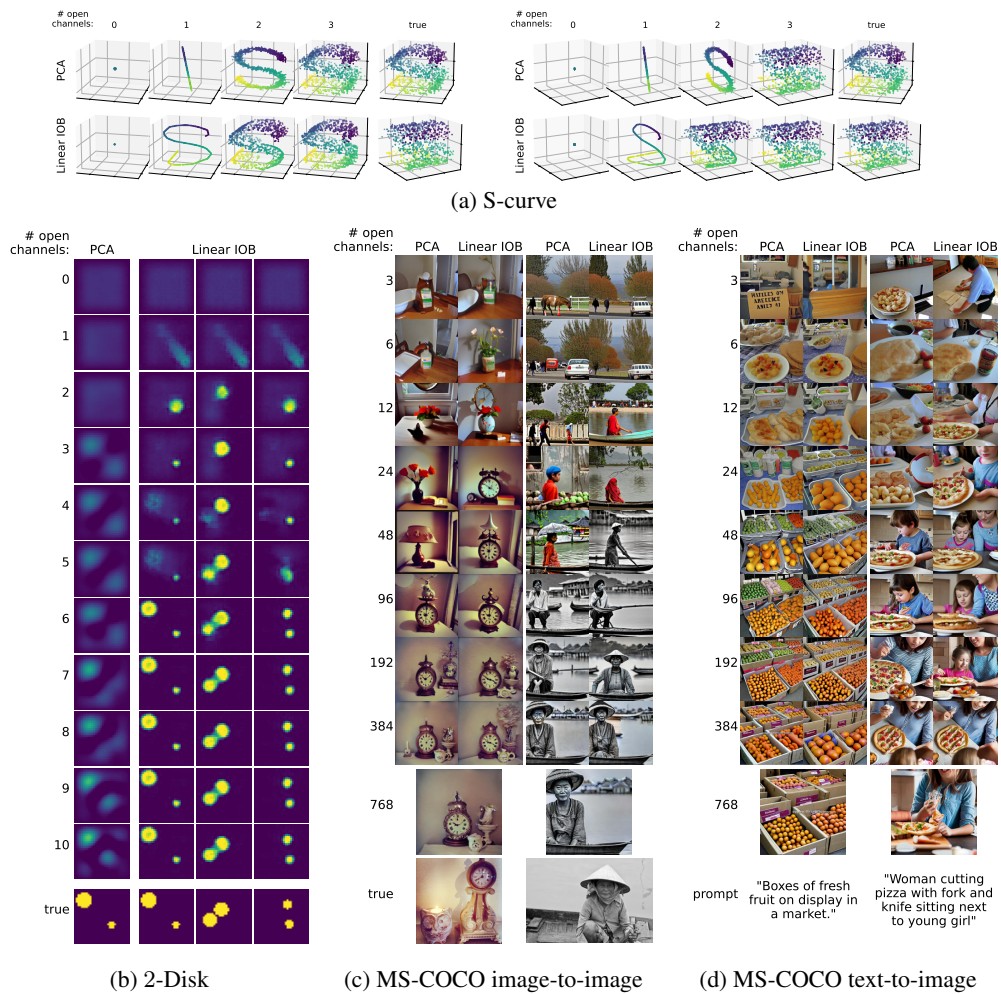

Figure 2: Reconstruction examples of various experiments as a function of bottleneck width after autoencoding compression with either PCA or Linear IOBs. For the MS-COCO experiments, images are generated using the same diffusion noise for all bottleneck widths, both for PCA and Linear IOB examples. Note, when the number of open channels is 768 for MS-COCO, there is no bottleneck compression for either PCA or IOB.

smaller, more specific features (e.g., backgrounds, clothing, secondary objects). We note that the embedding IOB models used in the MS-COCO example were trained with MSE minimization of CLIP latents. Therefore, they can recover semantic features of the fully-open unCLIP model (768 channels in Figure 3d) but not exactly those of the true image.

The learned IOB representations are akin to $k$-dimensional manifold-fitting. In the S-curve reconstruction, we first see the reconstructions falling on the mean point at $k = 0$, then tracing the S-line at $k = 1$, and finally matching the unbiased S-curve manifold (without noise) at $k = 2$. It is only when the problem is full rank (at $k = 3$) that the model attempts to recover noise properties in the full $\mathbb{R}^3$ space. Here, we note that although a sufficiently flexible neural network could represent the entire S-curve in a single latent, the IOBs here converge to the simpler solution of fitting a $k$-dimensional manifold as an approximation to the dataset.

Figure 3 shows the reconstruction performance on the full test set, plotting the mean reconstruction error as a function of bottleneck width for each of our IOB autoencoders and benchmark compressors. As expected, PCA and kernel PCA perform extremely poorly on most non-linear, high-dimensional datasets, except in the case of high-$k$ in the MS-COCO, wherein small variations in CLIP embeddings are poorly fit by the autoencoders. We observe that IOB models outperform

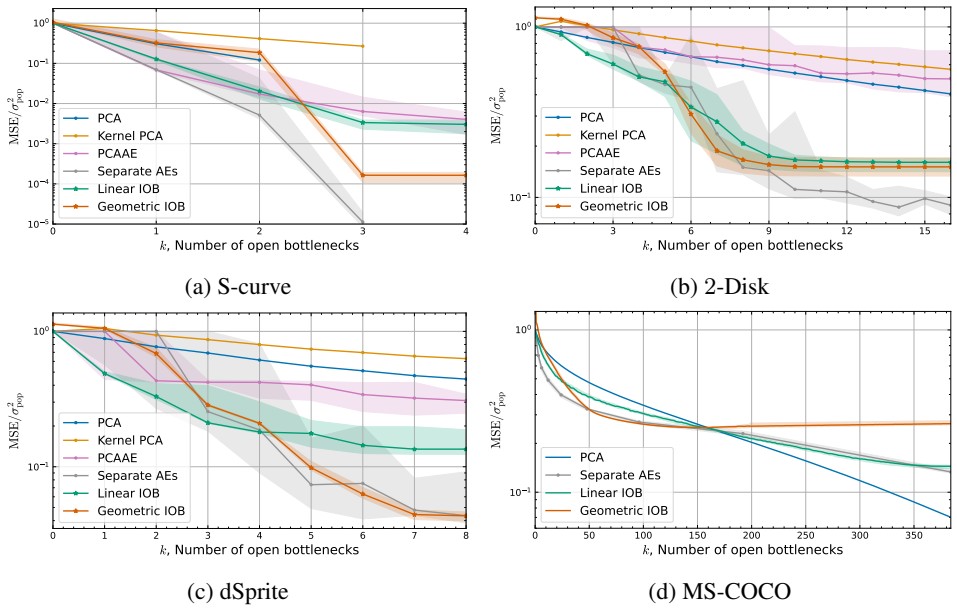

(a) S-curve

(b) 2-Disk

(c) dSprite

(d) MS-COCO

Figure 3: Median and 16-84th percentile mean-squared-error (MSE) reconstruction loss for different autoencoding compression schemes. All MSE's are normalized by the fixed population variance. Percentiles are computed from 10 independent runs of each configuration. For the S-curve dataset, the PCA loss goes to 0 at $k = 3$, and so is not shown in its plot. Also, the PCAAE model was not computed for the MS-COCO dataset, due to its intractability (See Section 5.1).

PCAAE reconstruction in almost all cases, a result also recorded in Staley & Markowitz (2022). The explicit latent disentanglement in PCAAE prevents the autoencoding scheme from learning embeddings of multiple latents at once, which is required for the S-curve, $n$-Disk, and dSprite datasets. The results also show that Linear and Geometric IOBs produce very similar compression for the S-curve, 2-Disk, and dSprite datasets but differ greatly in the high-$k$ regime of MS-COCO. This is caused by the fact that stochastic summation of Equation 2 is considerably less constraining than when it can be evaluated deterministically at full width. We then recommend that Linear IOBs be used for datasets under large encoding schemes.

Lastly, we observe that reconstructions of the IOB models roughly follow those of separate autoencoders and saturate at or around the intrinsic dimensionality of each dataset. For example, the S-curve reconstructions very quickly approach $1\%$ error at $k = 2$, the dimensionality of the S-manifold, and those of the 2-Disk experiment asymptote with $10\%$ reconstruction error at $k = 6 = 3n$ where $n = 2$ disks. This saturation is used to demonstrate ID estimates in Section 6. The fact that these models do not reach error $0$ is a natural limitation of our finite network architectures. We also note that, across many independent training runs, the reconstruction errors of both Linear and Geometric IOBs are very stable relative to those of separate autoencoders.

## 6 INTRINSIC DIMENSIONALITY ESTIMATION

Given the unique structure of the IOB, we can use nested model comparison to compare different bottleneck widths and construct estimates of global ID. The IOB layer $b_k : \mathbb{Z} \to \mathbb{Z}$ creates a bottleneck by masking the outputs of the $i$-th latent embeddings above $i > k$. We can consider this an element-wise multiplication, i.e. $g_k(\mathbf{z}) := \mathbf{z} \circ \mathbf{e}_k$, between the input latents $\mathbf{z}$ and a masking vector $\mathbf{e}_k := (\mathbb{1}_{k \geq 1}, \mathbb{1}_{k \geq 2}, \ldots, \mathbb{1}_{k \geq k_{\max}}) \in [0, 1]^{k_{\max}}$, where $\mathbb{1}$ is the indicator function. Next, we assume we have found optimal parameters $\theta^* = (\phi^*, \eta^*)$ following from Equation 2, allowing us to use our trained encoder $e_{\phi^*}$ to compress our test set inputs from $\mathcal{D}_{\text{test}} = \{\mathbf{x}_i, \mathbf{y}_i\}_{i=1}^{N_{\text{test}}}$ into a set of latents $\{\mathbf{z}_i\}_{i=1}^{N_{\text{test}}}$. Then, we define a composite log-likelihood, $\ell_{\mathbb{Z}} : \mathbb{Z} \to \mathbb{R}$, which takes as input the post-bottleneck latents and produces the log-likelihood of the test set in reconstruction space.

$$\ell_{\mathbb{Z}}(\mathbf{z}, \mathbf{y}) = \ell\left[d_{\eta^*}(\mathbf{z}), \mathbf{y}\right]. \tag{3}$$

| ID Estimator | S-curve | 1-Ball | 2-Ball | 3-Ball | 4-Ball | d-Sprites | MS-COCO |
|---|---|---|---|---|---|---|---|
| lPCA | 3.0 | 33.0 | 37.0 | 39.0 | 38.0 | 19.0 | 83.0 |
| MADA | 2.5 | - | 13.2 | 16.9 | 19.5 | 13.2 | 22.1 |
| TwoNN | 2.9 | 5.3 | 13.6 | 16.3 | 21.4 | 10.9 | 20.0 |
| TLE | 2.7 | 6.5 | 12.3 | 15.8 | 18.2 | 12.4 | 19.0 |
| CorrInt | 2.3 | 9.7 | 12.1 | 14.3 | 15.7 | 8.0 | 9.6 |
| MiND ML | 2.6 | 1.0 | 10.0 | **10.0** | 10.0 | 1.0 | 10.0 |
| MOM | 2.0 | 9.6 | 11.8 | 14.6 | 16.3 | 9.7 | 19.6 |
| MLE | 2.4 | 0.0 | 12.3 | 15.9 | 18.4 | 0.0 | 17.4 |
| FONDUE | - | - | - | - | - | 12.2 | - |
| Linear IOB* | **2** | **3** | **7** | 12 | **12** | **6** | 337 |
| Geometric IOB* | **2** | **3** | 8 | 12 | 13 | 7 | 361 |
| Data Dim. | 3 | 1024 | 1024 | 1024 | 1024 | 4096 | 768 |
| True Dim. | 2 | 3 | 6 | 9 | 12 | 5 | $\leq 768$ |

Table 1: Global ID estimates of synthetic and real datasets. In addition to the Linear and Geometric IOBs, we include estimates using lPCA (Cangelosi & Goriely, 2007), MADA (Farahmand et al., 2007), TwoNN (Facco et al., 2017), TLE (Amsaleg et al., 2018), CorrInt (Grassberger & Procaccia, 1983), MiND ML (Rozza et al., 2012), MOM Amsaleg et al. (2018), MLE (Haro et al., 2008), and FONDUE (Bonheme & Grzes, 2022). All baseline models apart from FONDUE were run using the implementations in Bac et al. (2021). Data dimensionality refers to the the dimensionality $\dim(\mathbb{X})$ of each sample in the training dataset, wheras true dimensionality refers to the number of tunable parameters used in the forward model of the dataset generation. Models introduced in this work are marked with an asterisk (*). ID estimates closest to the truth are bolded.

In essence, we have reframed our trained encoder-decoder network to evaluate the log-likelihood of test data in latent space. We then derive an estimator which maximizes the composite log-likelihood $\ell_{\mathbb{Z}}$ when given the pre-trained encoder's embeddings. We apply $g(\mathbf{z}; \mathbf{e}) = \mathbf{z} \circ \mathbf{e}$, a generalization of the linear $g_k$ bottleneck operation, but now with masking parameters $\mathbf{e}$ which can vary anywhere in real space $\mathbb{R}^{k_{\max}}$. Due to the training procedure of $d_{\eta^*}$, we would expect that the optimal $\mathbf{e}^* = \mathbf{1}$, where $\mathbf{1}$ is a vector of all ones. However, we can use the flexibility of these new variable parameters to quantify the improvement in $\ell_{\mathbb{Z}}$ as we allow more $\mathbf{e}$ components to vary.

We perform a likelihood ratio test to quantify the statistical significance of increasing the bottleneck width on the test data likelihood. Let us consider a null hypothesis $H_0$ in which $k$ bottleneck connections are open and compare it to an alternative hypothesis $H_1$ with $k+1$ connections open. Explicitly, we have that $H_0 : e_i = 0 \; \forall \; i > k$ and $H_1 : e_i = 0 \; \forall \; i > k + 1$. Following Wilks (1938), we expect that the distribution of twice the difference in the log-likelihood, i.e. $D = 2\left(\ell_{\mathbb{Z}}(H_1) - \ell_{\mathbb{Z}}(H_0)\right)$, will asymptote to a $\chi^2$ distribution for sufficient test data. Here, we use the notation $\ell_{\mathbb{Z}}(H_i)$ to represent the maximum log-likelihood $\ell_{\mathbb{Z}}$ averaged over the whole test set under hypothesis $H_i$. Using this test, we can examine these hypotheses down to a specified tolerance level (in our case $\alpha = 0.05$) to determine whether we need $k + 1$ parameters to describe the data. We then assign an ID to our dataset equal to the bottleneck width when we cannot reject the null hypothesis, i.e when we do not see sufficient gain in the log-likelihood for increasing $k$.

Table 1 compares the dimensionality estimates of our methods with various SOTA baselines. The performance of the estimators varies across the S-curve, $n$-Disk, and MS-COCO CLIP embedding datasets due to their different characteristics. While all estimators perform well on the S-curve dataset, differences arise in the $n$-Disk dataset, where the baselines estimate higher dimensionalities compared to the true dimensionality. However, IOB models provide closer estimates, often predicting to within one dimension of the truth. The IOB models generally overestimate the ID, possibly due to insufficient fitting or non-unique embeddings in the $n$-Disk datasets (i.e. when multiple balls either overlap or swap positions). Lastly, whereas the true dimensionality of MS-COCO embeddings in the CLIP semantic embedding space is unknown, we include ID estimates for completeness. Interestingly, the Linear IOB estimates an ID of 337, which is close to the 319-dimensional PCA compression that Ramesh et al. (2022) used to construct the unCLIP prior. We reserve an exhaustive comparison of the CLIP embedding dimensionalities for future work.

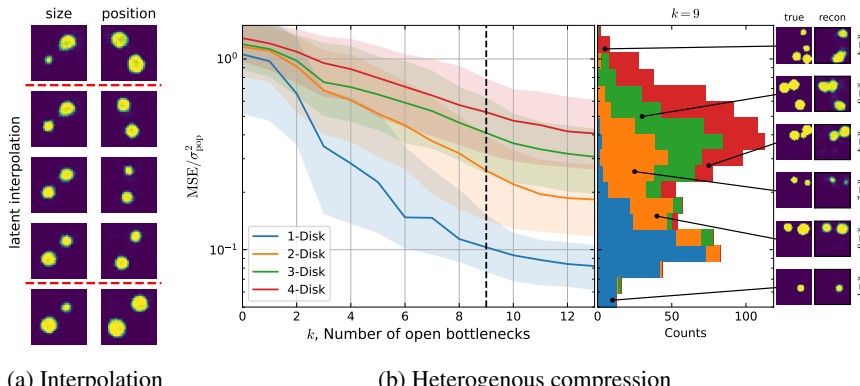

(a) Interpolation  (b) Heterogenous compression

Figure 4: IOB compression of a heterogenous dataset of $n$-Disk images where $n \in [1, 2, 3, 4]$. In **(a)**, reconstructions derived from encoding and decoding true samples are shown above the upper and below the lower red lines. All images between the red lines are derived from interpolating between these embeddings in the IOB latent space and decoding them into image space. In **(b)**, we show the median and 16-84th percentile intervals of each $n$-Disk population in a heterogenous compression. Example reconstructions on the right are drawn from the reconstructions at a bottleneck width of $k = 9$, and are shown with a black line corresponding to their position in the distribution and their true number of disks $n$ at the right.

# 7 DATA EXPLORATION

In Figure 4a, we demonstrate an example of interpolation within the latent space of a Linear IOB autoencoder trained on the 2-Disk dataset. By performing simple linear interpolation of the IOB latent variables, we can observe variations in the size or location of disks, resulting in the generation of novel images not present in the original training set. This suggests that the latent variables encode semantic information related to the parameters used for generating the 2-Disk samples. Such interpolation provides insights into the capabilities of the IOB model in capturing and manipulating underlying features within the data.

Figure 4b shows how training on a heterogenous dataset of mixed $n$-Disk examples from $n \in \{1, 2, 3, 4\}$ can lead to complexity rank-ordering and pattern discovery. At inference time, we clearly see a structured ordering of different image complexities with simpler images receiving lower MSE at inference time. Clearly, the least complex 1-Disk samples achieve optimal compression at earlier bottleneck widths, whereas more complex samples such as 3- or 4-Disks show much poorer compression under smaller bottlenecks. However, this scaling is not monotonic and we see considerable mixing among the $n$-Disk populations. This is the result of the behavior of disks in our images to partially or entirely overlap, reducing the effective number of disks $n$ by one. We demonstrate some of such examples at the right of Figure 4b. This test is akin to measurements of local ID at each data point, the specifics of which we plan to explore in future work.

# 8 CONCLUSIONS

In conclusion, this paper introduced a method for ordering latent variables based on their impact in minimizing the likelihood, offering a unified framework that incorporates multiple previous approaches. The results demonstrate the adaptive compression capabilities of the proposed method, achieving near-optimal data reconstruction within a given neural architecture while capturing semantic features in the inferred latent ordering. Additionally, the model proves effective in compressing high-dimensional data when used in conjunction with SOTA multimodal image-to-image and text-to-image models like unCLIP (Ramesh et al., 2022). Furthermore, a novel methodology for ID estimation is introduced, surpassing previous benchmarks through various synthetic experiments. For further details on limitations, please refer to Appendix A.

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
