## A    LIMITATIONS

In the proposed methods for ID estimation, we describe statistical tests of likelihood gain within the latent space of a pre-trained encoder-decoder. In practice, the ID we recover is highly dependent on the architecture and training limitations of the implemented neural networks. In this way, our chosen architectures act as hyperparameters to our dimensionality estimates. Although we expect that increasing the depth of the network would allow our dimensionality estimates to asymptote to some value, we have not explicitly tested this.

In addition, the unit sweeping procedure used to implement the geometric bottleneck does not produce an optimal fitting across all possible bottleneck widths and is dependent on the convergence criterium and the choice of geometric rate $r$. In our applications, we tried several rates $p$ which produced stable training, but did not fully explore the space of possible hyperparameters. We expect this to have an impact on the learned compressions as well as the recovered estimates of ID.

## B    SUPPLEMENTARY EXAMPLES

See Figures 5 and 6 for supplementary examples of MS-COCO image generation with IOB compression.

## C    IMPLEMENTATION DETAILS

### C.1    TRAINING PROCEDURE

All datasets undergo a $90\% - 10\%$ training-validation split which are fixed for all experiments. All figures and results shown in the manuscript are evaluated solely on the validation split. In all experiments, we use an Adam optimizer (Kingma & Ba, 2014) with a learning rate of $5 \times 10^{-5}$, no learning rate decay, and a batch size of 64. For all models except the Geometric IOB, we implement an early stopping criterion of a minimum of $0.01\%$ validation improvement over 20 epochs. For the Geometric IOB, we perform unit sweeping over every latent node, wherein our criterion for 'convergence' of each node is a minimum of $1\%$ validation improvement over 10 epochs. All neural network models are implemented in `Pytorch` (Paszke et al., 2019).

All experiments use a Gaussian log-likelihood with fixed variance for construction of the loss function in Equation 2. The variance is set to be the population variance, estimated empirically from the training set. In the case of the heterogenous compression for Figure 4b, $n$-Disk samples are treated as independent samples from the underlying data distribution and thus are considered to have the same population variance.

### C.2    EXPERIMENTS

The maximum width of bottleneck for each experiment were chosen to avoid computational intractability while maximizing the expected information gain in the bottleneck range. The architectures were chosen to mimic examples of previous deep fully-connected and convolutional autoencoders. However, we did not perform an exhaustive architecture search and leave this to future work.

For the S-curve dataset, we use fully-connected dense encoders and decoders. Including the input layer and the bottleneck, the sequence of nodes in each layer of the encoder is $3 - 64 - 64 - 4$. The sequence of decoder layers is simply the reverse of that of the encoder. All layers use a ReLU activation function. For the Geometric IOB of S-curve dataset, we use a rate of $p = 0.95$.

The generator for the $n$-Disk dataset is described explicitly in Algorithm 1. The autoencoder for this experiment is a convolutional neural network. The encoder $e_\phi$ is constructed from three convolution layers with respectively 4, 12, and 24 filters each of size $4 \times 4$, a stride of two, and edge padding of one. The convolutional layers are followed by three dense layers of width $256 - 128 - 16$, including the bottleneck of size $k_{\max} = 16$. The decoder $d_\eta$ has the reversed architecture of the encoder, except it uses convolutional upsampling layers instead of normal convolutional layers. All layers in

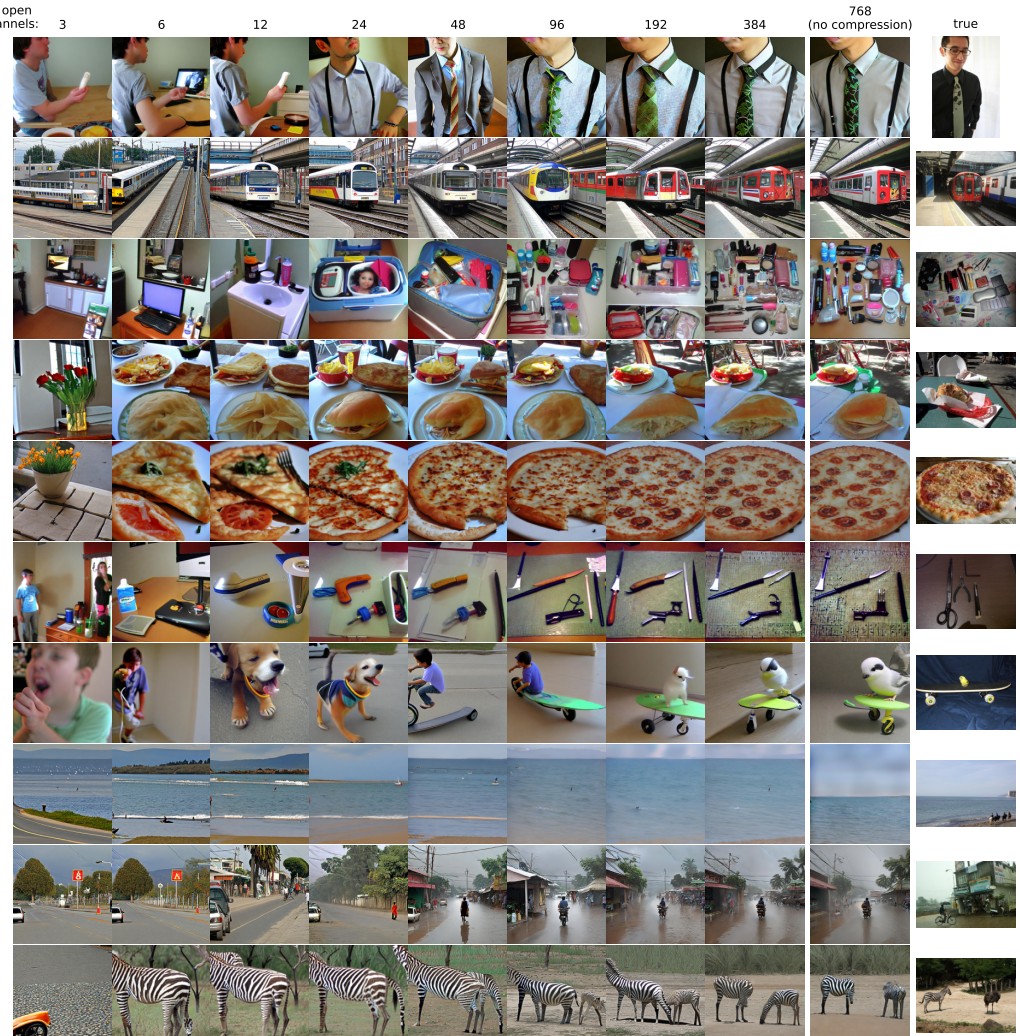

Figure 5: Supplementary examples of the MS-COCO image-to-image generation using latent compression with the Linear IOBs. Reconstructions are shown as a function of bottleneck width. Images are generated using the same diffusion noise for all bottleneck widths.

both the encoder and decoder use an ReLU activation function. For the Geometric IOB of all $n$-Disk datasets, we use a rate of $p = 1/3$.

For the dSprite dataset, we use a convolutional autoencoder as well. The encoder $e_\phi$ has four convolutional layers with respectively 64, 48, 48, and 24 filters each of size $4 \times 4$, a stride of two, and edge padding of one. These layers are followed by dense layers of width $384 - 256 - 128 - 8$, including the bottleneck of size $k_{\max} = 8$. As in the $n$-Disk dataset, the decoder has the reverse architecture of the encoder and all layers use a ReLU activation function. We also use a rate of $p = 1/3$ for dSprites.

Lastly, we use fully-connected dense encoders and decoders for the compression of CLIP embeddings of MS-COCO images. Including the input layer and the bottleneck, the sequence of nodes in each layer of the encoder is $768 - 384 - 384 - 384$. As in the S-curve experiment, the sequence of decoder layers is simply the reverse of that of the encoder and we use ReLU activation for each layer. For the Geometric IOB of the MS-COCO dataset, we use a rate of $p = 0.05$.

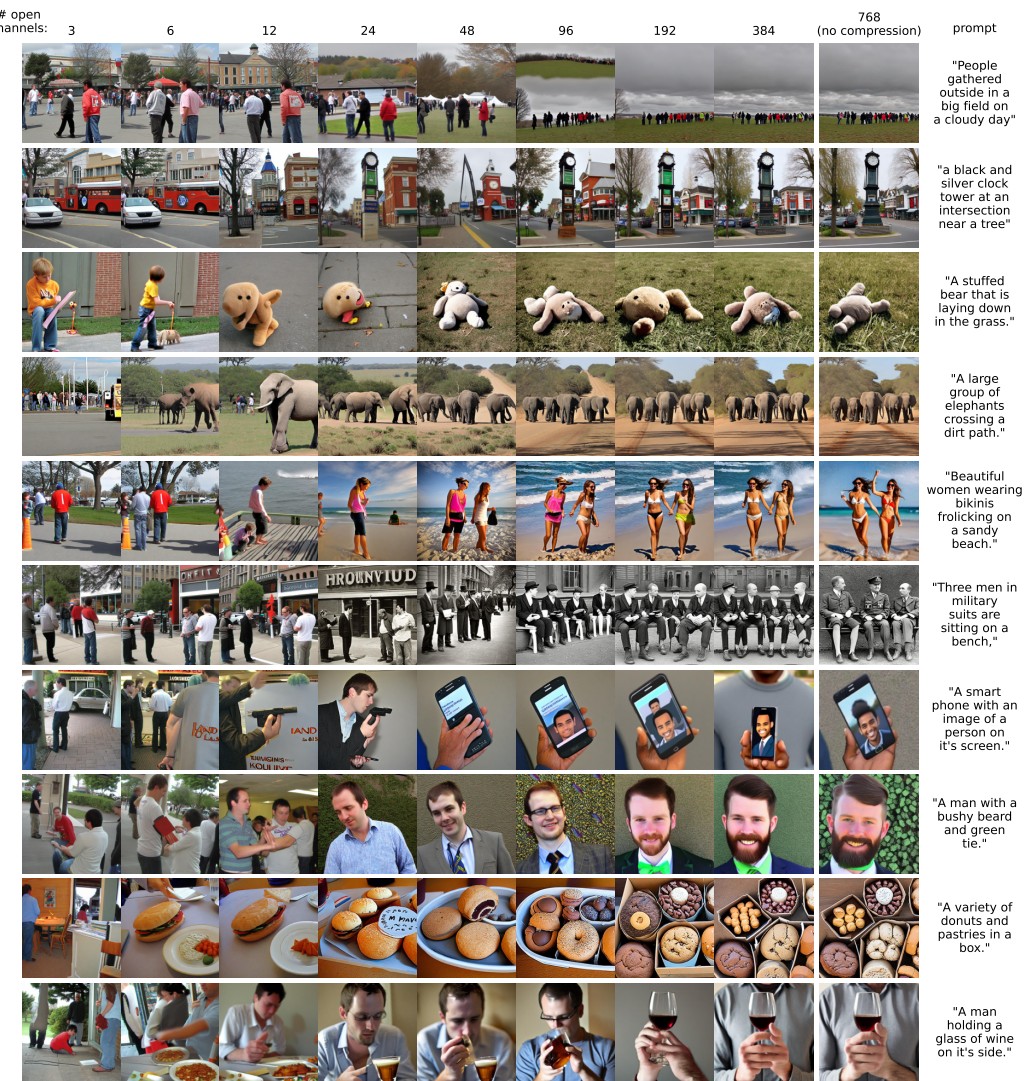

Figure 6: Supplementary examples of the MS-COCO text-to-image generation using latent compression with the Linear IOBs. Reconstructions are shown as a function of bottleneck width. Images are generated using the same diffusion noise for all bottleneck widths.

**Algorithm 1:** Synthetic generation of an image in the $n$-Disk datasets.

**Input**  : Number of disks $n$
**Output:** A single-channel image $\mathbf{x}$ of size $32 \times 32$ wherein $x_{ij} \in [0, 1] \ \forall \ i, j$

1  **for** $i \leftarrow 1$ *to* $n$ **do**
2  $\quad r_i \sim \mathcal{U}(2, 5)$
3  $\quad a_i \sim \mathcal{U}(r_i, 32 - r_i)$
4  $\quad b_i \sim \mathcal{U}(r_i, 32 - r_i)$
5  $\quad$ **for** $j \leftarrow 1$ *to* $32$ **do**
6  $\quad\quad$ **for** $k \leftarrow 1$ *to* $32$ **do**
7  $\quad\quad\quad$ **if** $r_i^2 \geq (a_i - j + 0.5)^2 + (b_i - k + 0.5)^2$ **then**
8  $\quad\quad\quad\quad x_{jk} \leftarrow 1;$
9  $\quad\quad\quad$ **else**
10 $\quad\quad\quad\quad x_{jk} \leftarrow 0;$
11 $\quad\quad\quad$ **end**
12 $\quad\quad$ **end**
13 $\quad$ **end**
14 **end**
15 **return** $\mathbf{x}$