# OpenReview forum: "Information-Ordered Bottlenecks for Adaptive Dimensionality Reduction"
_ICLR.cc/2024/Conference — Submitted to ICLR 2024_

### Official Review · Reviewer_mmf3 · 2023-10-31

**Soundness:** 2 fair
**Presentation:** 2 fair
**Contribution:** 2 fair
**Rating:** 5
**Confidence:** 3

**Summary:**

The paper presents a method for ordering latent variables based on their impact in minimising likelihood. The proposed framework generalises previous approaches. The adaptive compression capabilities of the model are demonstrated through experiments. The formalism also allows the intrinsic dimensionality problem to be addressed.

**Strengths:**

- Clearly defined scope
- Numerous experiments
- The method seems well suited for dimensionality reduction and shows promising results for intrinsic dimensionality estimation.

**Weaknesses:**

For the theoretical part
- The method depends on static hyperparameters: weights and k-max, with only two models of weights distribution proposed (geometric and linear) .

Regarding the experimental part :
- The organisation into several sections is confusing.
- Lack of numerical evaluation : it is difficult to compare or evaluate the performance of the method with the elements given in the experimental section (notably section 4.).
- Many toy data sets (S-curves, n-disk, dSprite)

**Questions:**

- Could the hyperparameters be learned or at least adjusted automatically ?
- Since the goal of compression is to somehow measure the amount of information, could any modification of the IOB layer provide a precise information-theoretic measure?

---

> ### Author Response · Authors · 2023-11-13
>
> Thank you for your review. We have addressed your questions and concerns in the responses below. We ask for further details on any confusing aspects of the organization, so that we may improve the quality of the work. We greatly value your feedback. We ask that you consider reevaluating the ratings in your report in light of our responses and proposed revisions.
>
> **The method depends on static hyperparameters: weights and k-max, with only two models of weights distribution proposed (geometric and linear).**
>
> The model is designed to be insensitive to $k_{\rm max}$. So long as $k_{\rm max}$ is sufficiently large, the bottleneck compression . This assumption is used and tested in Section 6, where we estimate intrinsic dimensionalities and show that the log-likelihood gain asymptotes at a fixed $k<k_{\rm max}$.
>
> We have also exhaustively tested different weighting schemes within the family of common statistical distributions (Uniform, Geometric, Poisson, Normal, etc.). We had found that IOB AE was largely insensitive to the form and hyperparameters. Instead, it was much more sensitive to the training procedure (i.e. stochastic vs deterministic summation of Eq 2). As a result, minding the length constraints, we did not include this testing in our main paper. However, if it should satisfy the reviewer's concern, we would be happy to include a section on hyperparameter tuning/sensitivity in the Appendix.
>
> **Could the hyperparameters be learned or at least adjusted automatically ?**
>
> Refer to the previous point.
>
> **Since the goal of compression is to somehow measure the amount of information...**
>
> This was done in one of the cited papers on Nested Dropout [1]. To avoid duplicating their work, we delibirately chose to focus this paper on intrinsic dimensionality estimation, integration with pre-trained models, and data exploration.
>
> However, we acknowledge that some language in the abstract regarding 'near-optimal compression' would beg some information-theoretic analysis. We are happy to change this in the revised manuscript if it should satisfy the reviewers concern.
>
> [1] Rippel et al. 2014 - https://arxiv.org/abs/1402.0915

---

> > ### Comment · Reviewer_mmf3 · 2023-11-23
> >
> > Thanks for the rebuttal.
> > I still think the paper needs to be revised for acceptance, especially in terms of experimental settings and comparisons.

---

### Official Review · Reviewer_RDmE · 2023-10-31

**Soundness:** 2 fair
**Presentation:** 2 fair
**Contribution:** 2 fair
**Rating:** 3
**Confidence:** 3

**Summary:**

This paper aims to seek an ordered latent embedding that can reveal its impact in minimizing the likelihood. In particular, the authors gradually increase the dimension of the latent embedding to ensure that top-ranked latent channels convey more impactful information. The proposed method has been demonstrated on synthetic and real image datasets using different pre-trained models.

**Strengths:**

1. The paper is overall easy to read, although some part in technology and experiment is not clear.
1. The study problem is interesting and the proposed method is very simple and intuitive.

**Weaknesses:**

1. Overall, the current version is insufficient and can not explain very well why the proposed method can deliver ordered latent variables.
1. It is claimed that the proposed method can be plugged into different pre-trained models, but there are only limited experiments from this perspective.
1. This paper lacks theoretical guidance and intuitive explanation behind the heuristic experiment settings.

**Questions:**

1. It is claimed in the abstract that this paper introduces a "novel theory for estimating global intrinsic dimensionality," but I cannot find any theoretical analysis related to this claim.
1. Regarding Eq. 2, it is unclear (i) whether $b_i$ is shared by all $f_{\theta}^i$ where $i\ge k$, and (ii) whether all the $b_i$ are updated at the same time.
1. In Fig. 2, the left panel of Fig. 2(a) is the same as the right panel of Fig. 2(a). In Fig. 2(c) and (d), it seems that the performance of PCA is similar to the proposed IOB. The baseline PCA is too simple; why not compare with more complex baselines, such as Kernel PCA and PCAAE?
1. Fig. 3 shows that Separate AEs are comparable with the proposed method. Does this mean that Separate AEs can perform the same function, i.e., delivering ordered latent embedding, as the proposed IOB?

---

> ### Author Response · Authors · 2023-11-13
>
> Thank you for your review. Below, we have addressed your listed weaknesses and questions. We hope we have addressed any miscommunication which might have affected understanding of the key aspects of our approach. We believe we have successfully addressed and clarified the reviewers key concerns. We ask that you consider reevaluating the ratings in your report in light of our responses and proposed revisions.
>
> **It is claimed in the abstract that this paper...**
>
> In this sentence, we are referring to our usage of Wilks Theorem and nested model comparison to establish an intrinsic dimensionality (ID) of our learned likelihood model. To our knowledge, this is the first application of this theory to bottlenecked autoencoders for the purpose of ID estimation. We acknowledge that the usage of the phrase 'novel theory' might be misleading, insinuating rigorous theoretical guarantees of ID convergence. Convergence of IDs is extremely difficult to show generally, as the definition of ID is inherently variable and  hyperparameter-dependent, and thus we do not provide evidence in this regard.
>
> We would be happy to change this term to 'novel methodology' from 'novel theory,' if this would satisfy the reviewer's concern.
>
> **Regarding Eq. 2, it is unclear (i) whether
>  is shared by..**
>
> We believe there is some significant confusion here. $b_i$ is not a variable but instead a deterministic function, without any tunable weights and, therefore, is not being updated at  all. $b_k(z)$ is our bottleneck operation and simply works by intaking a real-valued vector $z\in\mathbb{R}^n$ and outputting the same vector $z$, except all components $z_i=0$ for which $i>k$.
>
> **In Fig. 2, the left panel of Fig. 2(a) is the same as the right panel of Fig. 2(a).**
>
> This is done on purpose to show how the distribution looks in 3D from different angles. For example, the width of the S-curve in the Linear IOB with k=2 is difficult to observe on the left plots but is easy on the right. Similarly, the shape of the S at k=2 is hard to see on the right but easy on the left.
>
> **In Fig. 2(c) and (d), it seems that the performance of PCA is similar to the proposed IOB. The baseline PCA is too simple; why not compare with more complex baselines, such as Kernel PCA and PCAAE?**
>
> As described in the text, we clearly see key semantic features appearing at earlier bottleneck cutoffs in the IOB reconstruction than in the PCA reconstruction.  For example, in Fig 2c, the central clock is in the IOB reconstruction as early as $k=12$, but does not appear in the PCA reconstruction until $k=96$. In Fig 2d, the pizza and the child appear in the image at $k=24$ for IOB whereas they are evident in the PCA at $k=48$. Of course, these qualitative comparisons
>
> We did not have space to also show reconstructions of Kernel PCA and PCAAE in this figure. Inclusion of these baselines in the qualitative analysis would increase this figure size by a factor of two, and we are already reaching the page maximum. We chose to include PCA because it is well-known by the community and provides a clear intuitive comparison of the effect of IOB compression in Fig 2a and 2b. Quantitatively, kernel PCA and PCAAE only do marginally better than PCA (as shown in Fig 3), so these "more complex" baselines would show very little qualitative improvement.
>
> We are happy to add qualitative comparisons of kernel PCA and PCAAE in the appendix of a revised version of the manuscript, if this would satisfy the reviewers concern.
>
> **Fig. 3 shows that Separate AEs are comparable with..**
>
> As pointed out in the paper, separate autoencoders are only provided as a baseline for "optimal compression" given the neural architecture. By applying an IOB bottleneck, we are training a 'constrained' autoencoder, enforcing an ordered representation which is not strictly necessary for compression optimality. The purpose of including Seperate AEs is to show that this additional constraint doesn't significantly weaken our capacity for compression.
>
> The Separate AEs are strictly **not using ordered latent embeddings**. For example, an AE with $k=10$ and an AE with $k=11$ can have entirely distinct latent embeddings (in theory), because their weights are not shared. This violates the nested model comparison assumption necessary to make our ID estimates in Section 6. Also, it becomes extremely computationally expensive to train separate AEs for every possible bottleneck as the latent dimensionality increases. For our MS-COCO example, we would have needed to train 384 separate AEs to equal the performance of one IOB AE. This is another key advantage of our IOB approach, that we can get a computationally tractable ordered latent representation with this objective.

---

> > ### Comment · Reviewer_RDmE · 2023-11-23
> > **Thanks for the author's rebuttal.**
> >
> > Thanks for the author's rebuttal. It addresses some of my confusion. However, I am not convinced by the excuse of the lack of theoretical contribution and the 'improper' experimental setting due to the limited page space. More importantly, in my opinion, it is just a marginal technical contribution to claim a finding that has already been found and not claimed in previous studies. Therefore, i will keep my score.

---

### Official Review · Reviewer_nk9J · 2023-11-01

**Soundness:** 2 fair
**Presentation:** 2 fair
**Contribution:** 2 fair
**Rating:** 3
**Confidence:** 3

**Summary:**

This paper presents an information-ordered bottleneck~(IOB) layer to compress data into latent representations. The basis of IOB is dynamic adjustment bottleneck width at inference based on likelihood maximization. Given a neural architecture, IOB can compress the input data while maintaining the semantic properties of data. Experiments on synthetic and real datasets show the effectiveness of the designed IOB.

**Strengths:**

1. The research topic is interesting and important in DL community, that learning high-quality compressed representation.
2. The authors conduct many experiments on synthetic and real datasets to verify the effectiveness of the proposed IOB layer.

**Weaknesses:**

1. The motivation of this paper is not clear. The authors argue that the current compression methods, like PCA, and Auto-Encoder need dimension parameters artificially, however, the threshold k_max in IOB also depends on empirical.
2. Lacking of theoretical contribution. Directly masking top-k elements of latent variables seems too simple, and easy to discard semantic information. I hope the authors can provide analytic evidence of how to compress data without data distortion.
3. In experiments, why not compare current model compression methods, such as Distillation, Quantization, and Pruning?

**Questions:**

Refer weakness.

---

> ### Author Response · Authors · 2023-11-13
>
> Thank you very much for your review. We appreciate the further suggestions and concerns. We below provide clarification and suggest revisions to improve the quality of the work to match the reviewer's standards. We ask that you consider reevaluating the evaluations in your report in light of our responses and proposed revisions.
>
> **The motivation of this paper is not clear...**
>
> The IOB model is specifically designed to be insensitive to the choice of $k_{\rm max}$. As shown in Fig 3 and used as a key assumption in our ID estimation in Section 6, the IOB compression is expected to asymptote at a given $k<k_{\rm max}$ when we reach the ID of the dataset. So long as we set a $k_{\rm max}$ sufficiently large, the compression below this limit should reach an optimal configuration.
>
> We also note that the primary focuses of this paper is in ID estimation, integration with pre-trained models, and data exploration. The compression capacity of IOBs is simply a mechanism to achieve this end, which needed to be shown first. We acknowledge that some language in the abstract claiming 'near-optimal' compression might be misleading. This, of course, was meant with respect to Separate AEs, but could insinuate some further information-theoretic analysis of compression. We would be happy to tone the language down in the abstract and throughout the paper, if this would satisfy the reviewers concern.
>
> **Lacking of theoretical contribution...**
>
> Reiterating the previous point, the primary theoretical contribution is in proposing a new method for ID estimation and not in providng information-theoretic guarantees on compression capacity. We delibirately did not include analysis of information compression  because it was previously explored in the proposal of Nested Dropout [1] and we did not want to copy their analysis.
>
> We also insist that the simplicity of the solution is a strength and not a weakness, as it allows for easy implementation of IOBs and simple integration into existing SOTA pre-trained models.
>
> **In experiments, why not compare current model compression methods...**
>
> The goal of IOBs is not to develop a compressed model representation, but instead to develop an ordered compression scheme amenable to the experiments in ID estimation and data exploration in Sections 6 and 7. As a result, we did not consider model compression methods such as Distillation, Quantization, and Pruning. Respecting the page limit, we likely will not have room for this in a further revision of the manuscript. However it is certainly one of our priorities for further applications of IOBs in future applications.
>
> [1] Rippel et al. 2014 - https://arxiv.org/abs/1402.0915

---

### Official Review · Reviewer_E5s9 · 2023-11-05

**Soundness:** 3 good
**Presentation:** 3 good
**Contribution:** 2 fair
**Rating:** 5
**Confidence:** 5

**Summary:**

This paper introduces the Information-Ordered Bottleneck (IOB), that proposes to adaptively compresses data into hierarchically ordered latent variables, streamlining multiple existing methods into a unified framework. The IOB's flexibility is demonstrated through various synthetic and real datasets, including image and text embeddings, where it achieves near-optimal compression and meaningful latent ordering. Comparative results showcase IOB's superior performance in data reconstruction and intrinsic dimensionality estimation, outpacing state-of-the-art methods and facilitating exploratory dataset analysis.

**Strengths:**

* IOB provides an adaptive data compression mechanism, allowing for the dynamic adjustment of bottleneck width while ensuring likelihood maximization. This adaptability ensures the most crucial information is captured first, enhancing the efficiency of data utilization.
* IOB provides a unified framework putting other works such as the Nested Dropout, Triangular Dropout and PCAAE. Its application across diverse datasets, both synthetic and real, underscores its broad adaptability and relevance.
* IOB model's ability to capture semantically meaningful latent ordering and intrinsic dimensionality adds depth and interpretability to the learned latent space.

**Weaknesses:**

* The algorithmic novelty is not clear. The core of IOB formulation and training seems to be based on Nested Dropout and the unit-sweeping introduced in that work.
* Few key approaches on ordered dimension reduction have not been discussed, for example [1] where a stochastic latent dimension and ordered sparsity has been applied through a Bayesian prior, where IOB seems to be a special case of it where the weighting is fixed to be linear or geometric.


[1] Samaddar, Anirban, et al. "Sparsity-Inducing Categorical Prior Improves Robustness of the Information Bottleneck." International Conference on Artificial Intelligence and Statistics. PMLR, 2023.

**Questions:**

See the Weakness above

---

> ### Author Response · Authors · 2023-11-13
>
> Thank you very much for your informed review. See below our responses to the weaknesses. We ask that you consider reevaluating the evaluations in your report in light of our responses and proposed revisions.
>
> **The algorithmic novelty is not clear...**
>
> We acknowledge that the core of the approach has been considered in many ML subfields. The idea is, of course, a very simple extension of normal autoencoders (which we argue is a strength!), and thus it is unsuprising that others have considered this before. However, we remark that our progenitor papers (Nested Dropout, Triangular Dropout, SparC-IB) have never been cited together, and that this is the first paper providing a universal approach to this topic, unifying the previous methods.
>
> Also, we insist that our paper includes many novel elements that have never been explored in this IOB-like context, to the best of our knowledge. These include:
>  * Integration with pre-trained models, including the SOTA semantic, multimodal compression models. (Fig 2c and 2d).
>  * Intrinsic dimensionality (ID) estimation (Table 1)
>  * Latent embedding interpolation (Fig 4a)
>  * Heterogenous dataset interpretation through complexity ranking. (Fig 4b)
>
> These topics are the focus of our paper and differ greatly from the investigations in previous works (e.g. regularization and information compression analyses).
>
> **Few key approaches on ordered dimension reduction have not been discussed...**
>
> We thank the reviewer for making us aware of this previous work on Sparc-IB's. They are correct that our approach and this previous work share several properties, notably including a masked autoregressive prior on the latent vector. However, a key difference between the approaches is that for a given forward pass in Sparc-IB, the imposed bottleneck dimension $d_n$ is sampled from a categorical prior $\pi_{n,k}(X)$ which is a learned function of the input $X_n$. In our case, the categorical prior is fixed and the information pass-through is a continuous function of dimensionality. With its latent prior, Sparc-IB also includes elements of local ID estimation, although it is not thoroughly explored and benchmarked in their work, nor is it backed by statistical hypothesis testing, as in our approach.
>
> Our model could be cast as a Sparc-IB with specific choices of weighting distributions and priors and with a $d_n$ distribution which is purely dominated by a fixed prior.  The authors of Sparc-IB did not consider this configuration of their model, nor did they compare to previous approaches as in our study, so we insist that the analysis presented in this work remains valuable.
>
> We will certaintly include a reference and discussion of the Sparc-IB in the paper revision. We hope that this will satisfy the reviewers concern without discounting the qualtiy of our novel contributions (listed above). We believe an exhaustive comparison of our method and Sparc-IB is a crucial topic for follow-up work.

---

### Meta-Review · Area_Chair_3e6R · 2023-12-05

**Metareview:**

This paper presents and investigates the information-ordered bottleneck, which is a type of layer that can adaptively compress representations. While the reviewers did find the method and interpretability of results interesting, they generally found the contribution to lack sufficient technical novelty, given the landscape of prior work. Furthermore, concerns were raised about the experimental setup and comparisons.

**Justification For Why Not Higher Score:**

Lack of sufficient novelty and unconvincing experimental analysis.

**Justification For Why Not Lower Score:**

N/A

---

### Decision · Program_Chairs · 2024-01-16

Reject